Genes for degradation and utilization of uronic acid-containing polysaccharides of a marine bacterium Catenovulum sp. CCB-QB4

Furusawa Go furusawa@usm.my
Azami Nor Azura
Teh Aik-Hong
Centre for Chemical Biology, Universiti Sains Malaysia , Bayan Lepas , Penang , Malaysia
Silva Pedro
Electronic publication date: 2021 Mar 9
Publication date: 2021
Volume: 9
Electronic Location ID: e10929
Received 2020 Aug 5; Accepted 2021 Jan 20
Copyright: ©2021 Furusawa et al.
Copyright year: 2021
Copyright holder: Furusawa et al.
License: This is an open access article distributed under the terms of the Creative Commons Attribution License, which permits unrestricted use, distribution, reproduction and adaptation in any medium and for any purpose provided that it is properly attributed. For attribution, the original author(s), title, publication source (PeerJ) and either DOI or URL of the article must be cited.
License URL: https://creativecommons.org/licenses/by/4.0/

Keywords: Catenovulum, Uronic acid-containing polysaccharide, Polysaccharide lyase, Glycoside hydrolase, Marin bacteria

Funding: Universiti Sains Malaysia 304/PCCB/6315220 This project was financially supported by the short-term grant (304/PCCB/6315220) by Universiti Sains Malaysia. The funders had no role in study design, data collection and analysis, decision to publish, or preparation of the manuscript.

==============================
Background

Oligosaccharides from polysaccharides containing uronic acids are known to have many useful bioactivities. Thus, polysaccharide lyases (PLs) and glycoside hydrolases (GHs) involved in producing the oligosaccharides have attracted interest in both medical and industrial settings. The numerous polysaccharide lyases and glycoside hydrolases involved in producing the oligosaccharides were isolated from soil and marine microorganisms. Our previous report demonstrated that an agar-degrading bacterium, Catenovulum sp. CCB-QB4, isolated from a coastal area of Penang, Malaysia, possessed 183 glycoside hydrolases and 43 polysaccharide lyases in the genome. We expected that the strain might degrade and use uronic acid-containing polysaccharides as a carbon source, indicating that the strain has a potential for a source of novel genes for degrading the polysaccharides.

Methods

To confirm the expectation, the QB4 cells were cultured in artificial seawater media with uronic acid-containing polysaccharides, namely alginate, pectin (and saturated galacturonate), ulvan, and gellan gum, and the growth was observed. The genes involved in degradation and utilization of uronic acid-containing polysaccharides were explored in the QB4 genome using CAZy analysis and BlastP analysis.

Results

The QB4 cells were capable of using these polysaccharides as a carbon source, and especially, the cells exhibited a robust growth in the presence of alginate. 28 PLs and 22 GHs related to the degradation of these polysaccharides were found in the QB4 genome based on the CAZy database. Eleven polysaccharide lyases and 16 glycoside hydrolases contained lipobox motif, indicating that these enzymes play an important role in degrading the polysaccharides. Fourteen of 28 polysaccharide lyases were classified into ulvan lyase, and the QB4 genome possessed the most abundant ulvan lyase genes in the CAZy database. Besides, genes involved in uronic acid metabolisms were also present in the genome. These results were consistent with the cell growth. In the pectin metabolic pathway, the strain had genes for three different pathways. However, the growth experiment using saturated galacturonate exhibited that the strain can only use the pathway related to unsaturated galacturonate.

Introduction

Uronic acids are a class of sugar acids oxidized the hydroxyl group on C6 of aldoses. Uronic acids including D-glucuronic acid, D-galacturonic acid, D-mannuronic acid, L-guluronic acid, and L-iduronic acid are components of polysaccharides produced by animals (heparin), terrestrial plants (pectin), seaweed (alginate and ulvan), and bacteria (gellan gum) (De Lederkremer & Marino, 2003). In general, these polysaccharides such as pectin, alginate, ulvan, and gellan gum, have broad potential in many applications due to their excellent properties of biocompatibility, non-toxic, immunogenicity, availability, and relatively low cost (Morelli & Chiellini, 2010; Venkatesan et al., 2015; Rahman, Dafader & Banu, 2017).

Pectin is a polymer with a linear structure characterized by a backbone consisting of a few hundred to thousand D-galacturonic acid units linked together by α-(1 →4)-glycosidic linkages. It is found in the cell walls of the plant and intracellular layer of plant cells, mainly fruits, such as apples, oranges, and lemons (Mudgil, 2017). Pectin contains a significant amount of neutral sugar, typically L-rhamnose, L-arabinose, D-galactose, D-xylose, and D-glucose, which are linked to the hydroxyl groups on the number 2 and 3 carbons of the main chain. Pectin attracted attention due to its gelling capabilities (Penhasi & Meidan, 2015). Pectin is widely used as a thickener and stabilizing agent in the food industry (Munarin, Tanzi & Petrini, 2012).

Alginate, also known as alginic acid is an unbranched polymer composed of β-D-mannuronic acid (M) and α-L-guluronic acid (G), which are covalently (1-4)-liked. The residues are randomly arranged into MM-, GG- and MG-blocks (Meng & Liu, 2013). Alginate is distributed widely in the cell wall of marine brown agar and has long been used in the industry such as the medical field, fabric, food, and beverage industries as thickening, gel-forming, and colloidal stabilizing agents (Liakos et al., 2013; Martău, Mihai & Vodnar, 2019).

Ulvan is water-soluble polysaccharides found in the cell wall of green algae (Ulva and Enteromorpha) composed mainly of 3-sulfated rhamnose (Rha3S), glucuronic acid (GlcA), iduronic acid (IdoA), and xylose (Xyl) (Kim, Thomas & Li, 2011). Ulvan has attracted pharmaceutical and medical applications for its anti-viral, anti-coagulant, and anti-proliferative activities towards cancer cells and its immune-stimulating properties (Alves, Sousa & Reis, 2013). In addition, ulvan also showed to be an activator of plant defense and an inducer of plant resistance to fungal diseases (Alves, Sousa & Reis, 2013). Therefore, it is a good potential for agricultural applications.

Gellan gum is an exopolysaccharide produced from non-pathogenic, Gram-negative bacterium, Sphingomonas elodea (earlier Pseudomonas elodea) using aerobic fermentation (Vendrusculo, Pereira & Scamparini, 1994). Gellan gum consisting of repeating tetrasaccharide units of glucose, glucuronic acid, and rhamnose residues in a 2:1:1 ratio: [→3)- β-D-glucose-(1 →4)- β-D-glucuronic acid-(1 →4)- β-D-glucose-(1 →4)- α-L-rhamnose-(1 →] (Jansson, Lindberg & Sandford, 1983). Gellan gum has been used in medicine, pharmaceutical formulations, cosmetics, or tissue engineering. As a biocompatible polysaccharide, gellan gum is used in contact with or inside the body. Besides, gellan gum is also useful in the food and biotechnology industry as immobilization of enzymes and yeast cells (Iurciuc , Tincu)et al.(2015).

These polysaccharides could also be alternative sustainable sources for fermentative biofuel production (John et al., 2011). The PLs and GHs play an important role in the saccharification of the polysaccharides in the process of biofuel production (Edwards et al., 2011; Takeda et al., 2011; Li et al., 2015). For example, endo- and exolytic alginate lyases from Saccharophagus degradans A1 were co-displayed on the yeast cell surface, and the co-displaying yeasts were able to effectively produce monosaccharides (Takagi et al., 2016b). On the other hand, oligosaccharides from uronic acids generated by PLs are known to have multiple biological activities. For instance, alginate oligosaccharides stimulate the growth of human endothelial (Kawada et al., 1997) and keratinocytes cells (Kawada et al., 1999). In addition, the oligosaccharides also promote the growth and root elongation of rice and barley (Tomoda, Umemura & Adachi, 1994; Hien et al., 2000). Mandalari and colleagues reported that pectin oligosaccharides (POS) have probiotic effects through the improvement of bifidobacteria and lactobacillus (Mandalari et al., 2007). Besides, POS inhibited inflammation, fibrosis formation, as well as cancer progression, transformation, and metastasis (Bonnin, Garnier & Ralet, 2014). Thus, PLs and GHs have attracted considerable interest in both academic and commercial spheres.

Many bacterial species, such as the genera Agrobacterium, Bacillus, Cellulophaga, Clostridium, Erwinia, Flammeovirga, Flavobacterium, Microbulbifer, Pseudoalteromonas, Pseudomonas, Saccharophagus, Sphingomonas, Vibrio, Xanthomonas, and Zobellia, are known as bacteria that are capable of degrading uronic acid-containing polysaccharides, mainly alginate and pectin (Liu et al., 2019b; Dubey et al., 2016; Takagi et al., 2016a; Takagi et al., 2016b). The genus Catenovulum consisted of three species, C. agarivorans, C. maritimum, and C. sediminis, was also known to degrade agar (all three strains) and alginate (C. maritimum, and C. sediminis). Besides that, Catenovulum sp. LP was able to produce an ulvan lyase (Li et al., 2015; Qiao et al., 2020). However, the degradation and utilization pathway of uronic acid-containing polysaccharides containing alginate, gellan gum, pectin, and ulvan by the genus Catenovulum are poorly understood. Our group isolated Catenovulum sp. CCB-QB4 (referred to hereafter as QB4) from Queens Bay of Penang, Malaysia, and the complete genome sequence was reported (Lau et al., 2019). From the study, it was reported that the QB4 genome contained 183 GHs and 43 PLs. Based on the information, this bacterium is predicted to have the ability to utilize many polysaccharides. To confirm the expectation, in this study, the cell growth of QB4 in the presence of polysaccharides containing uronic acids, namely, alginate, pectin, ulvan, and gellan gum, was confirmed using a shake flask fermentation method. In addition, genes involved in degrading and utilizing these polysaccharides were explored in the QB4 genome. This study is the first report to describe the degradation and potential metabolic pathways for utilization of four different uronic acid-containing polysaccharides in the genus Catenovulum. 

Materials and Methods

Strain and chemicals

The QB4 cells (Lau et al., 2019) was cultured using high nutrient artificial seawater medium (H-ASWM) [0.5% tryptone, 2.4% (w/v) artificial sea salt mix (Marine Enterprises International), 10 mM HEPES, pH 7.6], as reported by Furusawa and co-workers (Furusawa et al., 2015). A total of 0.2% of polygalacturonic acid (pectic acid) (Nacalai Tesque), sodium alginate (Sigma-Aldrich), Gelzan™ CM (gellan gum) (Sigma-Aldrich), and ulvan were used as carbon source throughout the research.

Purification of ulvan

The ulvan was purified according to the method described by Tabarsa et al. (2012) with slight modification. 5 g of the dry powder of Ulva pertusa was dissolved in 100 mL of water and stirred at 65  °C for 3 h. The mixture was cooled and centrifuged at 10,000 g for 20 min at 15 °C using Sorvall™ RC 6 Plus Centrifuge (ThermoFisher Scientific). 4 volume of cold isopropanol was added into the supernatant to precipitate crude polysaccharide. The solution was left overnight at 4 °C. The crude ulvan was harvested by filtration, washed with 70% isopropanol several times, and dried overnight at 60 °C.

Determination of bacterial growth

The QB4 cells were inoculated in 10 mL of H-ASWM broth and cultured overnight at 30 °C. 0.1 mL of the cell suspension was inoculated into 100 mL of H-ASWM medium with 0.2% of each uronic acid as the carbon sources. Media with and without 0.2% glucose (Fisher Scientific) were used as positive and negative controls, respectively.

Each flask was inoculated with 0.1 mL of pre-cultured bacterial cell suspension and incubated at 30 °C with an agitation speed of 200 rpm on the orbital shaker. To monitor the bacterial growth, the colony-forming unit (CFU) counts were conducted by culturing the cells on H-ASWM agar plates because the optical density measurement was not suitable for the sample with gellan gum in which the broth was solidified by calcium ion contained in H-ASWM medium. 100 l of the cell suspension from each sample was collected every 3 h. The suspension was diluted into 900 l of H-ASWM medium. After that, sequential 10-fold serial dilutions were made, and 100l of aliquots of each dilution were plated on H-ASWM agar plates. Colonies were scored after incubation for 48 h at 30 °C, and the growth curve was constructed. The experiment was performed in triplicates. The generation time (G) was estimated by the method described by Aparna et al. based on the growth curve (Aparna, Parvathi & Kaniyassery, 2020). The number of generation (n) was determined by the method of n = (logb –logB)/log2, where B and b are the numbers of bacteria at the beginning of a time interval and at the end of the time interval, respectively. The generation time (G) was determined by the method of G = t/n, where t is the time interval (min).

To confirm the utilization of saturated galacturonate, the QB4 cells were culture in an H-ASWM medium with 0.2% saturated galacturonate. Media with and without 0.2% glucose was used as positive and negative controls, respectively. To monitor the bacterial growth, the optical density (OD600nm) was measured at 3 h intervals for 30 h using UV spectrophotometer UV-1800 (Shimadzu). The experiment was performed in triplicates.

Genomic analysis of Catenovulum sp. CCB-QB4

The complete genome of QB4 deposited at GenBank under the accession number CP026604–CP026605 was determined by Lau et al. (2019). Genes involved in the carbohydrate-active enzyme (CAZymes) in QB4 were predicted using dbCAN pipelines (Yin et al., 2012). Several enzymes related to uronic acid metabolism in QB4 were predicted by Blastp at the National Center for Biotechnology Information (NCBI) server (Bethesda, MD, U.S.A.) and were found using the Kyoto Encyclopedia of Genes and Genomes (KEGG) pathway database (Kanehisa & Goto, 2000). Finally, the amino acid sequence similarity of all enzymes was confirmed using Blastp analysis with Protein Data Base (PDB). Signal peptide prediction was conducted by LipoP 1.0 (Juncker et al., 2003) and SignalP 5.0 (Armenteros et al., 2019). For sequence alignment of DEH reductase, the amino acid sequence of A1-R and A1-R’ from Sphingomonas sp. A1 and DEH reductase from Saccharophagus degradans 2-40 were obtained from GenBank (https://www.ncbi.nlm.nih.gov/genbank/). The sequence alignment with 4 sequences was conducted by ClustalW (Thompson, Higgins & Gibson, 1994) at PRABI Lyon-Gerland (https://npsa-prabi.ibcp.fr/cgi-bin/npsa_automat.pl?page=/NPSA/npsa_clustalw.html).

Results

Growth confirmation of QB4 using uronic acids

To confirm the growth of QB4 in the presence of uronic acids, the cells of QB4 were cultured in H-ASWM with alginate, pectin, ulvan, and gellan gum. As shown in Fig. 1, the cells of QB4 showed good growth in the presence of the uronic acid-containing polysaccharides and glucose except for negative control. The cells of QB4 reached the early stationary phase at 9 h in H-ASWM with alginate and gellan gum. The generation time of alginate and gellan gum was 76.76 and 74.07 min. However, the cell number of the sample with gellan gum was 5.7 times lower than that of the sample with alginate. On the other hand, the cells cultured by pectin and glucose reached the early stationary phase at 18 h. In the case of using ulvan, the cells reached stationary phase at 24 h, and the sample exhibited the highest cell number at 30 h in the experiment. The generation time of the samples with glucose, pectin, and ulvan was 91.37, 95.74, and 85.10 min, which were longer than that of the samples with alginate and gellan gum. Thus, QB4 cells exhibited a robust growth in the presence of uronic acids. Subsequently, we focused on genes for degradation and utilization of uronic acids in the genome of QB4.

Figure 1 Growth confirmation of QB4 cultured with four different uronic acid-containing polysaccharides, alginate (Alg), pectin (Pct), Ulvan (Ulv), and gellan gum (Gel).

The growth was measured by cfu/mL. Media with and without 0.2% glucose was used as positive (Glu) and negative controls (NC), respectively. All data shown are mean values from three replicate experiments. Error bars denote the standard deviation of triplicate samples.

Genes for alginate degradation and utilization

Alginate is a major polysaccharide found in the cell wall of brown algae and consisted of guluronate and mannuronate arranged as 1,4-linked polysaccharides. First, alginate is degraded into oligomeric or monomeric units by alginate lyases. Eight alginate lyase genes were found in the QB4 genome (Table 1). Ad1-PL6, Ad2-PL6 and Ad3-PL6, Ad4-PL7, Ad5-PL7, Ad6-PL7, Ad7-PL7, and Ad8-PL17 were classified into family PL6, PL7, and PL17 based on the CAZy database, respectively. As a result of Blastp search with Protein Data Bank (PDB), Ad proteins were similar to Alygc from Paraglaciecola chathamensis (Ad1-PL6 and Ad3-PL6), AlyQ from Persicobacter sp. CCB-QB2 (Ad2-PL6, Ad4-PL7, and Ad5-PL7), alginate lyase of Klebsiella pneumoniae (Ad6-PL7 and Ad7-PL7), and Saccharophagus degradans 2-40 (Ad8-PL17) (Table S1). LipoP 1.0 (Juncker et al., 2003) was used to predict the signal peptides and their type of each enzyme. Table 1 demonstrated that Ad1-PL6, Ad2-PL6, Ad4-PL7and Ad5-PL7 possessed type I signal peptide, which was cleaved by signal peptidase I, and Ad6-PL7, Ad7-PL7, and Ad8-PL17 have type II lipoprotein signal peptide, which was cleaved by signal peptidase II. The lipoprotein signal peptide referred to as “lipobox” plays an important role in anchoring the protein on the outer surface of the cell membrane after secretion and modification of N-terminal cysteine residue (Pugsley, Chapon & Schwartz, 1986; Hutcheson, Zhang & Suvorov, 2011). This suggested that Ad6-PL7, Ad7-PL7, and Ad8-PL17 may localize on the cell surface. In contrast, Ad1-PL6, Ad2-PL6, Ad4-PL7, and Ad5-PL7 may be released into the culture medium. On the other hand, Ad3-PL6, which does not possess any signal peptides, may locate at the cytoplasm.

Table 1 Genes involving in alginate metabolism.

Alginate-degrading enzymes	
Abbreviation	Function	CAZy	Sp. GenBank	
Ad1_PL6	Alginate lyase	PL6	Type I	WP_108604939.1	
Ad2_PL6	Alginate lyase	PL6, CBM32, CBM32	Type I	WP_108601791.1	
Ad3_PL6	Alginate lyase	PL6, CBM16	-	WP_108605000.1	
Ad4_PL7	Alginate lyase	PL7, CBM32	Type I	WP_159084278.1	
Ad5_PL7	Alginate lyase	PL7, CBM32, CBM32	Type I	WP_108602212.1	
Ad6_PL7	Alginate lyase	PL7	Lipobox	WP_108602720.1	
Ad7_PL7	Alginate lyase	PL7	Lipobox	WP_108603319.1	
Ad8_PL17	Alginate lyase	PL17	Lipobox	WP_108601502.1	
Alginate-utilization proteins	
Abbreviation	Function	GenBank	
Au1	DEH reductase	WP_108601028.1	
Au2	KDG kinase	WP_108601027.1	
Au3	KDG kinase	WP_108603513.1	
Au4	2-dehydro-3-deoxyphosphogluconate aldolase	WP_108602314.1	
Au5	2-dehydro-3-deoxyphosphogluconate aldolase	WP_108603883.1	
Notes.

Sp. indicates signal peptide.

Takase et al. (2010) described that an exotype oligoalginate lyase, A1-IV, from Sphingomonas sp. A1 degraded oligoalginates into monosaccharides, which are then nonenzymatically converted to 4-deoxy-L-erythro-5-hexoseulose uronic acid (DEH). Takagi and colleagues also reported that an alginate lyase, Alg7K, from Saccharophagus degradans showed exolytic activity and produced monosaccharides from oligoalginates (Takagi et al., 2016a). The amino acid sequence of the alginate lyase, Ad6-PL7, showed high similarity (77.3%) to Alg7K, suggesting that Ad6-PL7 might mediate hydrolysis of oligoalginate to produce monomers. The DEH is converted to 2-keto-3-deoxy-D-gluconate (KDG) by NADH or NADPH-dependent DEH reductase (Takase et al., 2010; Takase et al., 2014; Kim et al., 2016). A result of the BLASTp search demonstrated that the SDR family oxidoreductase, Au1, found in the QB4 genome (Table 1) showed high similarity to NADH (A1-R’, 60.9%) and NADPH (A1-R, 50.8%)-dependent DEH reductase. In addition, the gene aliments of DEH reductases showed that the TGXXXGX motif and catalytic triad (Ser, Tyr, and Lys), which are highly conserved in SDR family enzymes (Takase et al., 2014), were conserved in the SDR family oxidoreductase (Fig. S1). These results indicated that Au1 might function as DEH reductase. KDG kinase catalyzes the conversion of KDG to 2-keto-3-deoxy-phosphogluconate (KDPG), and then KDGP is converted into D-glyceraldehyde-3-phosphate and pyruvate by 2-dehydro-3-deoxy-phosphogluconate aldolase via the Entner -Doudoroff pathway. Two of the genes encoding the KDG kinase, Au2 and Au3, and the aldolase, Au4, and Au5, were found in the genome of QB4 (Table 1). Finally, pyruvate produced by the pathway goes into further metabolic pathways for generating energy. The five Au proteins also demonstrated a high degree of amino acid sequence similarity (>50%) to DEH reductase A1-R’ from Sphingomonas sp. A1 (Au1), KDG kinase Shigella flexneri (Au2 and 3), 2-keto-3-deoxy-6-phosphogluconate aldolase from Thermotoga maritima (Au4), and KDGP aldolase from Escherichia coli (Au5) (Table S1). As mentioned above, QB4 possessed many alginate lyases and enzymes responsible for alginate utilization. This is consistent with the robust growth of QB4 in the presence of alginate, as shown in Fig. 1.

Genes for pectin degradation and utilization

As a first step, pectin is depolymerized by pectin lyases or polygalacturonases. Pectin and pectate lyases are classified into five families of PLs. Although pectin lyases attack highly methyl-esterified pectin, pectate lyases specifically attack non-methylated polygalacturonate or methylated pectin with a very low degree (Hugouvieux-Cotte-Pattat, Condemine & Shevchik, 2014). These enzymes degrade pectin to unsaturated pectic-oligosaccharides and disaccharides.

In the CAZy database, seven pectate lyases belonging to family PL1 (Pd1_PL1, Pd2_PL1, Pd3_PL1, Pd4_PL1, and Pd5_PL1), PL3 (Pd6_PL3), and PL10 (Pd7_PL10) family pectate lyases were found in the genome of QB4. The result of Blastp search using the PDB database on Pd proteins also demonstrated that these Pd proteins were similar (>45%) to pectate lyase (Pd1_PL1, Pd2_PL1, Pd3_PL1, Pd5_PL1, Pd6_PL3, and Pd7_PL10) and pectinesterase (Pd4_PL1) from other bacterial species (Table S2). CBM13 found in Pd1_PL1 and Pd6_PL3 and CBM35 found in Pd2_PL1 and Pd3_PL1 were capable of binding to multi-ligands, such as arabinan, arabinoxylan, and pectin (Fujimoto, 2013; Dhillon et al., 2018). However, CBM77 contained in Pd2_PL1 recognized homogalacturonan (Fujimoto, 2013). Pectin methylesterase (PME) domain from family 8 Carbohydrate Esterase (CE8) was found in Pd4_PL1 and Pd5_PL1. PMEs produce de-esterified homogalacturonan by catalyzing the de-esterification of the methoxyl group of the pectin; as a result, the products are effectively degraded by pectate lyases (Kashyap et al., 2001). Besides, although Pd1_PL1, Pd3_PL1, Pd4_PL1, and Pd7_PL10 possessed type I signal peptide, Pd5_PL1 and Pd6_PL3 had lipobox, indicating that both two lyases localize on the cell surface. Hence, these results suggested that Pd5_PL1 plays an important role in the pectin degradation of QB4 due to its de-esterification and depolymerization activities and its localization.

The unsaturated disaccharides generated by the pectate lyases are converted to 5-keto-4-deoxyuronate (DKI) by two distinct processes. One is that the disaccharides are degraded by oligogalacturonate lyase belonging to family PL22, and then the product, Δ-4,5-unsaturated galacturonate, is linearized into DKI by KdgF (Hobbs et al., 2016). The other is that unsaturated galacturonyl hydrolases belonging to GH105 degrades the disaccharides and directly releases DKI (Hobbs et al., 2019). In the QB4 genome, no PL22 enzymes were found in the genome. However, three GH105 proteins, Pd8_GH105, Pd9_GH105, and Pd10_GH105, were present in the genome (Table 2). These enzymes showed a high degree of similarity (49∼67%) on unsaturated rhamnogalacturonyl hydrolases (YteR) based on Blastp search with PDB database. These results indicated that oligogalacturonate might directly convert to DKI by GH105 proteins in QB4. DKI is converted to 2-keto-3-deoxygluconate (KDG) by two enzymes, a DKI isomerase (KduI) and a 2-dehydro-3-deoxy-D-gluconate 5-dehydrogenase (KduD) (Hobbs et al., 2019). The pathway analysis based on KEGG showed that four of KduI (Pu1, Pu2, Pu3, and Pu4) and KduD (Pu5, Pu6, Pu7, and Pu8) were present in the genome of QB4. Blastp search with the PDB database also exhibited that these enzymes showed a high degree of similarity (63∼71%) on KduD and KduI from Enterococcus faecalis, Escherichia coli, and a pectolytic bacterium, Pectobacterium carotovorum (Table S2). These results indicated that QB4 uses pectin as a carbon source for cell growth and is consistent with the result shown in Fig. 1.

Table 2 Genes involving in pectin metabolism.

Pectin-degrading enzymes	
Abbreviation	Function	CAZy	Sp.	GenBank	
Pd1_PL1	Pectate Lyase (Plasmid)	PL1, CBM13	Type I	WP_108605188.1	
Pd2_PL1	Pectate Lyase (Plasmid)	PL1, CBM35, CBM77	-	WP_108605291.1	
Pd3_PL1	Pectate Lyase (Plasmid)	PL1, CBM35	Type I	AWB69198	
Pd4_PL1	Pectate Lyase (Plasmid)	PL1, CE8	Type I	WP_159084287.1	
Pd5_PL1	Pectate Lyase (Plasmid)	PL1, CE8	Lipobox	WP_108605272.1	
Pd6_PL3	Pectate Lyase (Plasmid)	PL3, CBM13	Lipobox	WP_108605187.1	
Pd7_PL10	Pectate Lyase (Plasmid)	PL10	Type I	WP_108605295.1	
Pd8_GH105	Unsaturated galacturonyl hydrolases (Plasmid)	GH105	Lipobox	WP_108605228.1	
Pd9_GH105	Unsaturated galacturonyl hydrolases (Plasmid)	GH105	Type I	WP_108605230.1	
Pd10_GH105	Unsaturated galacturonyl hydrolases (Plasmid)	GH105	Lipobox	WP_108605292.1	
Pd11_GH28	Polygalacturonases (Plasmid)	GH28	Tat	WP_108605238.1 ]	
Pd12_GH28	Polygalacturonases (Plasmid)	GH28	Type I	WP_108605277.1	
Pectin-utilization proteins	
Abbreviation	Function	GenBank	
Pu1	5-dehydro-4-deoxy-D-glucuronate isomerase (KduI)	WP_108601704.1	
Pu2	5-dehydro-4-deoxy-D-glucuronate isomerase (KduI)	WP_108601929.1	
Pu3	5-dehydro-4-deoxy-D-glucuronate isomerase (KduI)	WP_108602166.1	
Pu4	5-dehydro-4-deoxy-D-glucuronate isomerase (KduI) (Plasmid)	WP_108605242.1	
Pu5	2-dehydro-3-deoxy-D-gluconate 5-dehydrogenase (KduD)	WP_108601158.1	
Pu6	2-dehydro-3-deoxy-D-gluconate 5-dehydrogenase (KduD)	WP_108601705.1	
Pu7	2-dehydro-3-deoxy-D-gluconate 5-dehydrogenase (KduD)	WP_108602165.1	
Pu8	2-dehydro-3-deoxy-D-gluconate 5-dehydrogenase (KduD) (Plasmid)	WP_108605243.1	
Pu9	Glucuronate isomerase	WP_108601474.1	
Pu10	Glucuronate isomerase	WP_108604964.1	
Pu11	Glucuronate isomerase (Plasmid)	WP_108605248.1	
Pu12	Tagaturonate reductase	WP_108601943.1	
Pu13	Altronate hydrolase	WP_108601944.1	
Pu14	Tagaturonate epimerase (Plasmid)	WP_108605252.1	
Pu15	Fructuronate reductase	WP_108602164.1	
Pu16	Mannoate dehydratese	WP_108601469.1	
Notes.

Plasmid indicated that the gene is present in plasmid, not the genome.

The pathway analysis also showed the other pathway using saturated galacturonate (monosaccharide) produced by polygalacturonases. As mentioned above, although QB4 possesses seven pectate lyases, only two polygalacturonases (GH28), Pd11_GH28 and Pd12_GH28, which were highly similar (>66%) to polygalacturonase from pectolytic bacteria, such as Erwinia carotovora and Thermotoga marítima (Table S2), were found in the genome. Based on SignalP 5.0 analysis, it was found that Pd11_GH28 and Pd12_GH28 contained Tat and type I signal peptide, respectively. Saturated galacturonate is converted to D-tagaturonate by glucuronate isomerase. Three glucuronate isomerases, Pu9, Pu10, and Pu11, which were highly similar (> %) to glucuronate isomerase of Salmonella enterica subsp. enterica serovar Typhimurium and uronate isomerase of Caulobacter vibrioides CB15 (Table S2), were found in the QB4 genome. The conversion of D-tagaturonate to KDG occurs through two distinctly different pathways. One is that the process consists of two steps involving tagaturonate reductase and altronate hydrolase and one intermediate, D-altronate (Richard & Hilditch, 2009). The other is that tagaturonate is converted via three steps involving tagaturonate epimerase, fructuronate reductase, and mannonate dehydratase and two intermediates, D-fructuronate and D-mannonate (Valk et al., 2020). As shown in Table 2, QB4 possessed all genes involved in the two different pathways, namely tagaturonate reductase (Pu12), altronate hydrolase (Pu13), tagaturonate epimerase (Pu14), fructuronate reductase (Pu15), and mannoate dehydratase (Pu16). Homologous enzymes of these proteins were also found in the PDB database (Table S2). This result suggested that QB4 is also capable of using saturated galacturonate as a carbon source. In order to confirm this suggestion, the QB4 cells were cultured in H-ASWM broth with saturated galacturonate. As shown in Fig. 2, although the cells with glucose demonstrated robust growth, the cells were unable to grow in the broth with saturated galacturonate as well as the negative control in the incubation period. This result indicated that the unsaturated galacturonate utilization pathway is the main pathway to utilize polygalacturonic acid of QB4. Interestingly, genes for all pectate lyases, polygalacturonases, and one set of KduI (Pu4), KduD (Pu8), glucuronate isomerase (Pu11), and tagaturonate reductase were located in a plasmid (https://www.ncbi.nlm.nih.gov/nuccore/NZ_CP026605.1) found in QB4 (Table 2).

Genes for ulvan degradation and utilization

Ulvan lyases classified into 5 families, PL24, PL25, PL28, PL 37, and PL40, in the CAZy database (Li et al., 2020) were isolated from several marine bacteria, such as genera, Alteromonas, Pseudoalteromonas, Formosa, and Nonlabens (Kopel et al., 2016; Qin et al., 2018; Ulaganathan et al., 2018a; Ulaganathan et al., 2018b; Reisky et al., 2019). Ulvan lyases cleave between L-rhamnose 3-sulfate (Rha3S) and D-glucuronic acid (GlcA) or L-iduronic acid (IdoA). Fourteen ulvan lyases in QB4 were classified into three PLs families, PL24 (Ud1_PL24, Ud2_PL24, Ud3_PL24, Ud4_PL24, Ud5_PL24, Ud6_PL24, Ud7_PL24, Ud8_PL24, and Ud9_PL24), PL25 (Ud10_PL25, Ud11_PL25, Ud12_PL25, and Ud13_PL25) and PL40 (Ud14_PL40). Ud proteins belonging to the PL24 family were highly similar (>62%) to short ulvan lyase of Alteromonas sp. LOR, and Ud proteins belonging to the PL25 family were highly similar (>57%) to ulvan lyase-PL25 of Pseudoalteromonas sp. PLSV (Table S3). The number of the ulvan lyases was higher than that of Formosa agariphila (three PL40 proteins) (Reisky et al., 2019) and Alteromonas sp. LOR (one PL24 and one nonclassified protein) (Foran et al., 2017). Ud6_PL24 and Ud9_PL24 and the enzymes belonging to PL25 possessed lipobox. Although Ud2_PL24, Ud3_PL24, Ud4_PL24, Ud5_PL24, and Ud8_PL24 contained type I signal peptide, the remaining enzymes did not have any signal peptides. Although only one pectin lyase had a lipobox, many ulvan lyases possessed lipobox as well as alginate lyases, indicating that alginate and ulvan degradation activity of QB4 may be more effective than pectin degradation of the strain.

Figure 2 Growth confirmation of QB4 cultured with saturated galacturonate (Sat. Gal).

The growth was measured as optical density (OD600 nm). Media with and without 0.2% glucose was used as positive (Glu) and negative controls (NC), respectively. All data shown are mean values from three replicate experiments. Error bars denote the standard deviation of triplicate samples.

The unsaturated uronyl residue at the non-reducing end of the oligosaccharides produced by the ulvan lyases may be released by unsaturated glucuronyl hydrolases (GH105). Five putative unsaturated glucuronyl hydrolases, Ud15_GH105, Ud16_GH105, Ud17_GH105, Ud18_GH105, and Ud19_GH105 were found in the QB4 genome and showed similarity on unsaturated beta-glucuronyl hydrolase of Nonlabens ulvanivorans, which is known as a ulvan-degrading bacterium, based on BLASTp search (Table 3; Table S3) (Collén et al., 2014). After forming DKI by the enzymes, the following process may be the same as pectin utilization.

In the ulvan-degrading process of F. agariphila, Rha3S-Xyl-Rha3S was also the main product by ulvan lyases (Reisky et al., 2019). First, Rha3S at the non-reducing end is desulfated by sulfatases. The amino acid sequences of two sulfatases, Ulu1 and Ulu2, in the QB4 genome showed high similarity (77.3 and 86.2%) to the sulfatases, WP.032096151.1 and WP.632096147.1, located into ulvan utilization loci of Alteromonas sp. LOR (Foran et al., 2017), indicating that the two sulfatases may involve desulfation of the Rha3S. The Rha3 will remove the Rha3-Xyl-Rha3S by α-L-rhamnosidase. Five α-L-rhamnosidases classified into GH78 family, Ud20_GH78, Ud21_GH78, Ud22_GH78, Ud23_GH78, and Ud24_GH78, were found in the QB4 genome and showed similarity on rhamnosidase from Bacillus sp. GL1, Dictyoglomus thermophilum, Streptomyces avermitilis, and Bacteroides thetaiotaomicron (Table S3). Ud21_GH78, Ud22_GH78, Ud23_GH78, and Ud24_GH78 contained lipobox, suggesting that these enzymes are located on the cell surface. In addition, Ud20_GH78 and Ud22_GH78 possessed the CBM67 domain, which binds L-rhamnose in a calcium-dependent manner (Fujimoto et al., 2013), suggesting that the two α-L-rhamnosidases are the main components to generate Rha3.

Rha3 metabolic pathway was described by Reisky et al. (2019). α-L-rhamnose is converted to β-L-rhamnose by L-rhamnose mutarotase. Next, isomerization of the β-L-rhamnose to L-rhamnulose is catalyzed by rhamnose isomerase. The product is converted to L-rhamnulose-1-phosphate by pentulose/hexulose kinase (rhamnulokinase), and subsequently, the L-rhamnulose-1-phosphate is cleaved into L-lactaldehyde and dihydroxyacetone phosphate by rhamnulose aldolase. Finally, the L-lactaldehyde is converted to pyruvate by aldehyde dehydrogenase and lactate dehydrogenase. Table 3 displayed that the corresponding genes (Ulu3, Ulu4, Ulu5, Ulu6, Ulu7, and Ulu8) were found in the QB4 genome, and these proteins showed similarity on corresponding proteins in the PDB database (Table S3). This result indicated that Rha3 is metabolized by QB4 cells. The genes involving the metabolic pathway downstream of the mutarotation in F. agariphila form a gene cluster (Reisky et al., 2019). However, the genes in QB4 were randomly distributed in the genome.

Xyl will be released by β-xylosidase classified into GH3 and GH43 families. Based on the CAZy database, seven GH43 (Ud25_GH43, Ud26_GH43, Ud27_GH43, Ud28_GH43, Ud29_GH43, Ud30_GH43, and Ud31_GH43) proteins were found in the QB1 genome while F. agariphila possessed two each of GH3 and GH43 proteins (Reisky et al., 2019). Ud25_GH43, Ud29_GH43, and Ud31_GH43 were similar to glycoside hydrolases from Zobellia galactanivorans (50.00%) and Halothermothrix orenii H 168 (45.75 and 39.10%) (Table S3). The D-xylose is converted to D-xylulose-5-phosphate via xylose isomerase and xylulose kinase, and then the product is passed to the pentose phosphate pathway. Two genes encoding xylose isomerase (Ulu9) and xylulose kinase (Ulu10) were found in the QB4 genome based on RAST server annotation and formed a gene cluster with transcription repressor, xylR (Ulu11) (Table 3). These results suggested that QB4 cells may use not only glucuronic acid but also L-rhamnose and D-xylose as carbon sources. In addition, the presence of numerous genes encoding ulvan lyase may promote strong ulvan degradation and its utilization. The QB4 cells indeed exhibited a robust cell growth in the presence of ulvan as well as that in the presence of alginate, as shown in Fig. 1.

Genes for gellan gum degradation and utilization

Gellan gum is depolymerized gellan lyases classified into PL33 family proteins. Gellan lyases are found in several bacterial species, such as Bacillus sp. GL1, Geobacillus stearothermophilus 98, and Opitutaceae bacterium TAV5 (Hashimoto et al., 1997; Derekova et al., 2006; Helbert et al., 2019). Gellan gum is degraded to tetrasaccharides composed of [→3)- β-D-glucose-(1 →4)- β-D-glucuronic acid-(1 →4)- β-D-glucose-(1 →4)- α-L-rhamnose-(1 →] by gellan lyases. The tetrasaccharides are completely degraded to monosaccharides by β-D-glucosidase (GH1, GH2, GH3, GH5, GH9, GH16, GH30, GH39, and GH116), unsaturated glucuronyl hydrolase (GH88 and GH105), and α-L-rhamnosidase (GH78) (Hashimoto et al., 2003).

Table 3 Genes involving in ulvan metabolism.

Ulvan-degrading enzymes	
Abbreviation	Function	CAZy	Sp.	GenBank	
Ud1_PL24	Ulvan lyase	PL24		WP_108601530.1	
Ud2_PL24	Ulvan lyase	PL24	Type I	WP_108601531.1	
Ud3_PL24	Ulvan lyase	PL24	Type I	WP_108604943.1	
Ud4_PL24	Ulvan lyase	PL24	Type I	WP_108601549.1	
Ud5_PL24	Ulvan lyase	PL24	Type I	WP_108601554.1	
Ud6_PL24	Ulvan lyase	PL24	Lipobox	WP_108602230.1	
Ud7_PL24	Ulvan lyase	PL24, CBM32		WP_108604992.1	
Ud8_PL24	Ulvan lyase	PL24	Type I	WP_108602276.1	
Ud9_PL24	Ulvan lyase	PL24	Lipobox	WP_108602290.1	
Ud10_PL25	Ulvan lyase	PL25	Lipobox	WP_108601547.1	
Ud11_PL25	Ulvan lyase	PL25	Lipobox	WP_108601636.1	
Ud12_PL25	Ulvan lyase	PL25	Lipobox	WP_108601685.1	
Ud13_PL25	Ulvan lyase	PL25	Lipobox	WP_108602265.1	
Ud14_PL40	Ulvan lyase	PL40		WP_108602237.1	
Ud15_GH105	Unsaturated glucuronyl hydrolase	GH105	Lipobox	WP_108601620.1	
Ud16_GH105	Unsaturated glucuronyl hydrolase	GH105	Lipobox	WP_108601696.1	
Ud17_GH105	Unsaturated glucuronyl hydrolase	GH105	Lipobox	WP_108601928.1	
Ud18_GH105	Unsaturated glucuronyl hydrolase	GH105	Lipobox	WP_108602225.1	
Ud19_GH105	Unsaturated glucuronyl hydrolase	GH105	Lipobox	WP_108602277.1	
Ud20_GH78	α-L-rhamnosidase	GH78, CBM67	Type I	WP_108601540.1	
Ud21_GH78	α-L-rhamnosidase	GH78	Lipobox	WP_108604942.1	
Ud22_GH78	α-L-rhamnosidase	GH78, CBM67	Lipobox	WP_108601635.1	
Ud23_GH78	α-L-rhamnosidase	GH78	Lipobox	WP_108602268.1	
Ud24_GH78	α-L-rhamnosidase	GH78	Lipobox	WP_108602275.1	
Ud25_GH43	Putative β-xylosidase	GH43	Type I	WP_108601365.1	
Ud26_GH43	Putative β-xylosidase	GH43	Lipobox	WP_108601561.1	
Ud27_GH43	β-xylosidase	GH43	Type I	WP_108601631.1	
Ud28_GH43	β-xylosidase	GH43	Lipobox	WP_108602133.1	
Ud29_GH43	β-xylosidase	GH43	Lipobox	WP_159084088.1	
Ud30_GH43	β-xylosidase	GH43	Lipobox	WP_108602377.1	
Ud31_GH43	β-xylosidase	GH43	Lipobox	WP_108602397.1	
Ulvan-utilization proteins	
Abbreviation	Function	GenBank	
Ulu1	Sulfatase	WP_108601682.1	
Ulu2	Sulfatase	WP_108601697.1	
Ulu3	L-rhamnose mutarotase	WP_108601622.1	
Ulu4	Rhamnose isomerase	WP_108601699.1	
Ulu5	Pentulose/hexulose kinase (rhamnulokinase)	WP_108602148.1	
Ulu6	Rhamnulose aldolase	WP_108604932.1	
Ulu7	Aldehyde dehydrogenase	WP_108601159.1	
Ulu8	Lactate dehydrogenase	WP_108602249.1	
Ulu9	Xylose isomerase	WP_108604489.1	
Ulu10	Xylulose kinase	WP_108604490.1	
Ulu11	Transcriptional regulatory protein XylR	WP_10860449.1	

Figure 1 demonstrated that QB4 might also use gellan gum as a carbon source for its growth. However, gellan lyases were not found in the genome. As shown in Table 3, QB4 possessed five unsaturated glucuronyl hydrolases and five α-L-rhamnosidases (Table 3). The hydrolysates, DEH, and α-L-rhamnose, may be metabolized by the pathway described above. In addition, one β-D-glucosidase classified into the GH1 family (Gd1-GH1) was also found in the genome, and Gd1-GH1 showed high similarity (63.23%) on β-glucosidase A of Hungateiclostridium thermocellum in the PDB database (Table 4; Table S4). β-D-glucose produced by the β-D-glucosidase is converted to D-glucose-6-phosphate by glucokinase, and then the product is converted to β-D-fructose-6-phosphate, which is an intermediate of glycolysis, by glucose-6-phosphate isomerase. One of these genes, Gu1 and Gu2, were found in the QB4 genome (Table 4) and were similar on glucokinase and glucose-6-phosphate Isomerase of E. coli in PDB database, respectively (Table S4), indicating that β-D-glucose is converted to β-D-fructose-6-phosphate and is metabolized by glycolysis.

Discussion

Polysaccharides from plants, seaweed, and bacteria, especially their oligosaccharides, have attracted considerable attention due to various biological activities. Thus, polysaccharide lyases and glycosyl hydrolases were isolated from microorganisms, including fungi and bacteria.

The genus Catenovulum have been known as bacteria that can degrade agar and polysaccharide containing uronic acids, such as alginate, pectin, and ulvan (Cui et al., 2014; Li et al., 2015; Lee, Lee & Hong, 2019; Liu et al., 2019a; Xie et al., 2013). Our previous report demonstrated that many genes encoding polysaccharide-degrading enzymes were present in the QB4 genome (Lau et al., 2019). In this study, expectedly, QB4 cells were able to degrade and used the polysaccharides as a carbon source for their growth (Fig. 1). Especially, the QB4 cells exhibited a robust growth in the presence of alginate and ulvan, suggesting that polysaccharides from marine algae are effectively used by QB4 cells rather than that of plants and bacteria. The growth profile of QB4 was different from S. degradans of which the cells reached the early stationary phase at 9 h with alginate, glucose, and pectin (Takagi et al., 2016a).

Table 4 Genes involving in ulvan metabolism.

Gellan-degrading enzymes	
Abbreviation	Function	CAZy	Sp.	GenBank	
Gd1_GH1	β-D-glucosidase	GH1		WP_159084202.1	
Gellan-utilization proteins	
Abbreviation	Function	GenBank	
Gu1	Glucokinase	WP_108603884.1	
Gu2	Glucose-6-phosphate isomerase	WP_108604264.1	

Three pathways of pectin metabolisms were known in bacteria. One is the polygalacturonate pathway metabolized unsaturated disaccharides. The pathway was found in bacteria, such as Escherichia coli and phytopathogenic enterobacterium, Erwinia chrysanthemi (Chatterjee, Thurn & Tyrell, 1985; Richard & Hilditch, 2009). The other two pathways metabolize saturated galacturonate through uronate isomerase (isomerase pathway) (Richard & Hilditch, 2009) or tagaturonate epimerase (epimerase pathway) (Rodionova et al., 2012). Table 2 displayed that the QB4 possessed genes involved in the three pathways. However, QB4 cells did not grow in the presence of saturated galacturonate (Fig. 2). Although a pectin-degrading marine bacterium, Pseudoalteromonas sp. PS47, also has genes for the epimerase pathway, the strain was unable to grow on saturated galacturonate (Hobbs et al., 2019). Hobbs and colleagues proposed that GH28s of the strain are periplasmic proteins and, thus, that saturated galacturonate would be produced in the periplasm (Hobbs et al., 2019). It was suggested that the strain might be unable to transport extracellular saturated galacturonate. On the other hand, although Pd11_GH28 containing Tat signal peptide would be exported into periplasm through Tat (for twin-arginine translocation) system (Stanley, Palmer & Berks, 2000), Pd12_GH28 would be secreted to the outside of the cells via Sec system, suggesting that saturated galacturonate might be generated in the culture broth in contrast to the strain PS47. However, the QB4 did not grow using saturated galacturonate. Thus, it seems like the BQ4 cells are also unable to transport extracellular saturated galacturonate as well as the strain PS47. In other cases, some investigators reported that the isomerase pathway is not crucial for utilizing pectin in Dickeya dadantii (formerly E. chrysanthemi) (Hugouvieux-Cotte-Pattat et al., 1996; Pédron et al., 2018). It was known that D. dadantii belonging to the class Gammaproteobacteria, which is a plant pathogen and pectinolytic bacterium, also possessed the polygalacturonate pathway and the isomerase pathway (Hugouvieux-Cotte-Pattat et al., 1996). Even though the enzyme production of the isomerase pathway was not impaired by a mutation in genes of the polygalacturonate pathway (kduD and kduI), these mutants did not exhibit their growth on polygalacturonate (Hugouvieux-Cotte-Pattat et al., 1996). In addition, the transcriptomic analysis of D. dadantii during the early colonization on the plant leaf demonstrated that genes involved in the polygalacturonate pathway were upregulated in the condition (Pédron et al., 2018). These results suggested that the isomerase pathway was not the main pathway for utilizing pectin in the bacterium. The third pathway, epimerase pathway, involving the conversion of D-tagatose to D-fructuronate by tagatose the epimerase (UxaE), was found in the hyperthermophilic bacterium, Thermotoga maritima, belonging to the phylum Thermotoga. However, the pathway was not found in other bacteria, such as Escherichia and Bacillus (Kuivanen et al., 2019). uxaE and other epimerase pathway genes, such as hexuronate catabolism regulator (uxaR), fructuronate reductase (uxaD), predicted D-mannonate utilization enzyme (gntE) constituted a regulon regulated by the GntR-like transcription factor UxaR in the genus Thermotoga (Rodionova et al., 2012). A regulon is a gene cluster or operon that is regulated by the same regulatory protein. However, Table 2 demonstrated that genes encoding Pu14, Pu15, and Pu16 were scattered in the chromosome and plasmid, suggesting that Pu14, Pu15, and Pu16 might not constitute a regulon. In addition, Fig. 2 demonstrated that the saturated galacturonate, which is a substrate of the epimerase pathway, was not used by QB4 cells. These results suggested that the epimerase pathway might not function in QB4.

Figure 1 demonstrated that the QB4 cells exhibited a robust growth using ulvan as a carbon source. Qiao et al. (2020) fermented Catenovulum sp. LP using the shaking-flask method containing 1.2% purified ulvan, of which the concentration was 6 times-higher than that of the QB4. The bacterial culture reached the stationary phase at 36 h incubation period (Qiao et al., 2020) that was relatively slower than that of the QB4. Table S5 showed that the QB4 genome contained 14 ulvan lyase genes, which is the largest number compared to Siansivirga zeaxanthinifaciens CC-SAMT-1 (8 genes) Tamlana sp. UJ94 (7 genes), Polaribacter sp. BM10 (7 genes), and Wenyingzhuangia fucanilytica CZ1127 (7 genes) in the CAZy database (Table S5). The numerous genes in the QB4 genome may be crucial to efficiently degrade ulvan and to stimulate robust growth.

Figure 1 demonstrated that the QB4 cells were capable of using gellan gum as a carbon source, however, no gallan lyases were found in the QB4 genome. It was known that Bacillus sp. GL1 and G. stearothermophilus 98, which possessed all genes for gellan gum degradation and utilization, were able to use gellan gum as a sole carbon source (Hashimoto et al., 1998; Derekova et al., 2006). On the other hand, although Paludisphaera borealis PX4 belonging to the order Planctomycetales does not have any gellan lyases, the bacterium was capable of degrading gellan gum (Ivanova et al., 2017). The authors suggested that an unsaturated glucuronyl hydrolase and two α-L-rhamnosidases of the bacterium involve degrading gellan gum. Tables 3 and 4 showed that QB4 cells possess five unsaturated glucuronyl hydrolases, five α-L-rhamnosidases, and one β-D-glucosidase. We expected that these hydrolases play a critical role in degrading gellan gum in QB4 cells. The lack of gellan lyases may cause lower degradation efficiency. Thus, the cell growth with gellan gum was not robust compared to those with other uronic acids.

Conclusion

The agar-degrading bacterium, Catenovulum sp. CCB-QB4, used uronic acids, including alginate, pectin ulvan, and gellan gum, as carbon sources for its growth. Especially, QB4 cells exhibited robust cell growth in the presence of alginate and ulvan from seaweed. In gene analysis based on the CAZy database, a large number of polysaccharide lyases and hydrolases involved in degrading these uronic acids were found in the QB4 genome. Many alginate lyases, ulvan lyases, unsaturated glucuronyl hydrolases, α-L-rhamnosidase, and β-xylosidase contained lipobox, indicating that QB4 cells can effectively degrade alginate and ulvan and uptake the oligosaccharides. Of course, genes for metabolizing the uronic acids were also present in the QB4. These results were suggested that QB4 will become a source of novel uronic acid degradation enzymes.

Supplemental Information

Supplemental Information 1 Sequence alignment of homology regions of DFH reductase

Red boxes indicate a TGXXXGX motif. Asterisks indicate catalytic triads of the enzyme. Sa.d., Saccharophagus degradans; Sp. Sphingomonas sp. The figure was drawn using the program ESPript (Robert and Gouet, 2014).

Reference: Robert, X. and Gouet, P. (2014) ”Deciphering key features in protein structures with the new ENDscript server”. Nucleic. Acids Research 42 (W1), W320-W324 - doi: 10.1093/nar/gku316.

Click here for additional data file.

Supplemental Information 2 The similarities of amino acid (aa) sequence of alginate metabolic enzymes of QB4 in PDB database

Click here for additional data file.

Supplemental Information 3 The similarities of amino acid (aa) sequence of pectin metabolic enzymes of QB4 in PDB database

Click here for additional data file.

Supplemental Information 4 The similarities of amino acid (aa) sequence of ulvan metabolic enzymes of QB4 in PDB database

Click here for additional data file.

Supplemental Information 5 The similarities of amino acid (aa) sequence of ulvan metabolic enzymes of QB4 in PDB database

Click here for additional data file.

Supplemental Information 6 List of bacterial species possessing 5 or more ulvan lyases in CAZy database

Click here for additional data file.

Supplemental Information 7 Fig. 1 raw data

Click here for additional data file.

Supplemental Information 8 Fig. 2 raw data

Click here for additional data file.

Additional Information and Declarations

Competing Interests

Author Contributions

Data Availability

The authors declare there are no competing interests.

Go Furusawa conceived and designed the experiments, performed the experiments, analyzed the data, prepared figures and/or tables, authored or reviewed drafts of the paper, and approved the final draft.

Nor Azura Azami performed the experiments, analyzed the data, authored or reviewed drafts of the paper, and approved the final draft.

Aik-Hong Teh conceived and designed the experiments, authored or reviewed drafts of the paper, and approved the final draft.

The following information was supplied regarding data availability:

Raw data is available in the Supplemental Files.

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
