# Peer review of "Genes for degradation and utilization of uronic acid-containing polysaccharides of a marine bacterium Catenovulum sp. CCB-QB4"

_PeerJ, doi:10.7717/peerj.10929_

## Round 0.1 · original submission · Major Revisions

Please ensure all comments are addressed and that any speculation is clearly labelled as such and kept to a minimum.

Reviewer 1 ·

Basic reporting

The article makes an interesting analysis based on genome mining and is has been written in correct English
In the background, please clearly indicate in rows 36 and 37 the abbreviation of polysaccharide lyases (PLs) and glycoside hydrolases (GHs) to take into account in subsequent sections.
Since the main objective of the article is to verify the ability of the bacteria to degrade polysaccharides based on the presence of enzyme genes, the introduction should include a concise review of the corresponding Cazy PLs y GHs enzyme families. As it stands, it has only provided information on the structure of the polysaccharides used as a carbon source and the application of their corresponding oligosaccharides.
The structure of the article follows the standard format.
Regarding the figures, I believe that figure 1 is sufficient and it is necessary to graph it on a semilog scale.
Figure 2 does not have relevant data to show and may only be indicated in the text, in addition to that a different method of growth measurement has been used.
Files S3 and S4 of supplementary material should have clear titles and a brief description of the data displayed.
In file 4 an incomplete graph is observed.
Since the hypothesis is supported mainly by bioinformatic analysis, the other data referred to growth on polysaccharides based media (figure 1) should be strengthened, including, for example, a kinetic analysis with quantitative parameters (growth rate µ h-1) (carried out by applying basic mathematical modelling).

Experimental design

The article highlights the ability of Catenovulum sp. CCB-QB4 to degrade polysaccharides (alginate, pectin ulvan, and gellan gum) and generate oligosaccharides with different applications.
The hypothesis deduces that the presence of genes for degradation and utilization of polysaccharides (alginate, pectin ulvan, and gellan gum) in the genome of CCB-QB4 as well as its ability to grow in minimal media with these compunds as a carbon source can provide evidence of active metabolic pathways. Give that transcriptomics approaches offers direct evidence related to expression of the genes involved in the degradation of these polysaccharides, it is necessary to justify it because the proposed methodology may be sufficient to support the working hypothesis.
About the microorganism, Catenovulum sp. CCB-QB4, since some pectin utilization genes are present in the plasmid of the bacterium, indicate whether in all cases it has been verified that the bacterium has maintained its plasmid in all experiments.
Also please clarify if these genes are exclusive to the plasmid or are also present in the genome (this is not indicated in the Lau et a. 2019 reference).
About M&M: Please explain why two different growth measurement methods have been chosen. The data in figure 1 have been obtained by plate counting, while those in figure 2 result from measuring OD at 600nm. In both figures (1 and 2) the shape of the growth curve on glucose as well as the growth times are different.

Validity of the findings

The article has a simple experimental approach, but the bioinformatic analysis contributes positively to the relevance of its results.
The conclusions are adequate although in some way future studies should be suggested for contribute to elucidate the metabolic potential of CCB-QB4.

Additional comments

The article is simple but interesting and can contribute to the knowledge about the metabolic capacities of CCB-QB4.

Reviewer 2 ·

Basic reporting

The manuscript investigated the growth conditions under different uronic acid-containing polysaccharides, sequenced and annotated its genomic information. However, it is not suitable for publication in Peer J due to its preliminary experiment and insufficient works. It should be considered to be submitted to more professional journal such as Marine Genomics et al.

Experimental design

The experiment is preliminary and the results are insufficient to support the research.

Validity of the findings

The validity of the findings is sound and it need more results like protein expression and characterization to support the validity of the findings.

Additional comments

The manuscript investigated the growth conditions under different uronic acid-containing polysaccharides, sequenced and annotated its genomic information. However, it is not suitable for publication in Peer J due to its preliminary experiment and insufficient works. It should be considered to be submitted to more professional journal such as Marine Genomics et al.

Reviewer 3 ·

Basic reporting

The manuscript provided by Furusawa et al. describes the characterization of genes involved in uronic acid degradation within the bacterium catenovulum strains QB4. In its current state, I felt that the manuscript was not focused making it hard to identify the main objective of this research, and several important data required to justify their claims were missing.

First of all, it is unclear overall on what the authors wish to convey to the readers. The authors presented the data by stating that strain QB4 harbors numerous genes involved in the depolymerization and utilization of uronic acids but did not state clearly on whether they wanted to say that QB4 is a strain that is unique and the first of its kind in comparison to currently reported strains or whether it is only among the catenovulum genus. Currently, based on the objectives and discussions provided, the authors wish to convey the later message but if this is so, the introduction at its current state is not appropriate. If they wish to convey the later message, the introduction should focus on catenovulum strains including a review on the use or ability of catenovulum strains in relation to uronic acid or polysaccharide degradation. If they wish to convey the former message, it is recommended that the authors provided a wider review covering the major bacterial strains in relation to uronic acid and polysaccharide degradation. Either way, it is recommended that the authors clarify the objective of their research and provide an introduction that supports the significance of their research. Please comment and if necessary, amend accordingly.

Experimental design

In addition to the overall message of the manuscript, several points also need to be addressed regarding the experimental design.

The authors provided data on the degradation of the various polysaccharides in Figure 1 and 2 by presenting the growth of strain QB4 in relation to each polysaccharide, but did not provide data quantitating the reduction or accumulation of the polysaccharides nor performed assays to identify the end-products of each utilized polysaccharide. It is hard to speculate on what genes or enzymes are being used by strain QB4 without this data. Why did the authors exclude these analyses as these data would allow readers to better understand the characteristics of strain QB4, its capability in uronic acid degradation, or what are the possible products that could be attained using this bacterium. Please comment. If there was a reason why they excluded the end-product assays and analyses, they should explain this in the manuscript as it is common in enzymology for researchers to perform analyses on the end-products of a given enzyme.

The authors also should include information on the sequence similarity (percentage) of their enzymes with those identified from databases in Table 1-3 as this would allow readers to determine the novelty of the proteins within QB4. Proteins that may have been annotated to a functional protein does not necessarily mean that the identified protein does function as the similar protein. It all depends on the similarity of these proteins and the presence of motifs and enzyme activity sites. Therefore, it is necessary for the authors to provide this information as it will support whether the discussion and speculations made by the authors are true or not. Please provide the necessary data.

Validity of the findings

In terms of the validity of the findings, in the results section, assuming that the authors did determine the function of their annotated genes based on the function regardless of the percentage of sequence similarity, the authors provided a unique way of describing their findings by speculating the role of strain QB4 where they used information of enzymes reported by other groups. This was observed in Line 200 – 214 and throughout the Results section of the manuscript. It is really dangerous to come up with such speculation without supporting experimental data. Terms like “indicated” were also used but again this is commonly used when referring to data or results attained from experiments. Please avoid presenting speculations or terms without supporting data. The authors should also not mix up the results and discussion together as they are meant to be separated. Please comment and amend accordingly.

Additional comments

Line 285, please amend “Qb4” to “QB4”.

---

## Round 0.2 · Major Revisions

I agree with the need for additional characterization of the reaction products (requested earlier by reviewer #3) . I also need some more data regarding the conclusions taken from Fig.2: authors state "As shown in Figure 2, although the cells with glucose demonstrated a robust growth, the cells with were unable to grow in the broth with saturated galacturonate as well as negative control in the incubation period. This indicated that the unsaturated galacturonate utilization pathway is the main pathway to utilize polygalacturonic acid of QB4." I do not think that the lack of utilization of saturated galacturonate means that unsaturated galacturonate is used instead: one could envision an enzyme that degraded uronic acid PS into dissacharaides and (e.g.) the saturated digalacturonatetherein obtained could be further transformed/epimerized/tec before being cleaved. That would allow growth from those PS without the ability to use either saturated or unsaturated galacturonate. Maybe the authors mean, instead, that all pathways known so far used either saturated or unsaturated galacturonate, and that the lack of growth on saturated galacturonate strongly suggests that unsaturated galacturonate is used instead, but in that case, they should both be quite clear regarding all known pathways (preferably including detailed reaction schemes of the known metabolic pathways) and stress that their conclusions come from a process of hypothesis elimination, rather than confirmation of the hypothesis

Reviewer 1 ·

Basic reporting

No comment

Experimental design

No comment

Validity of the findings

I think this sentence: "This study is first report to describe the degradation and utilization pathway of four different uronic acid-containing polysaccharides in the genus Catenovulum" is not fully supported by experimental data and are partially based on bioinformatic analysis.

Then it is suggested to replace as follow:

This study is first report to describe the degradation and probable metabolic pathways for utilization of four different uronic acid-containing polysaccharides in the genus Catenovulum.

Additional comments

The introduction of the manuscript has been improved according to the recommendations made previously and more references have been included. Only one last suggestion has been included for the final paragraph.
The methods have been explained more clearly and some experimental calculations have been included that better analysis of the growth curves.
Figure 1 has also been corrected and the organization and presentation of supplementary material has been improved.
There are no more suggestions to improve the manuscript.

Reviewer 3 ·

Basic reporting

Having reevaluated the contents of the revised manuscript, the authors have improved the introduction to suit the objective of their research. Therefore I feel that the manuscript is now sound and readability has improved.

Data was requested in relevance to the assay of the end-products upon polysaccharide degradation (Fig.1 and Fig. 2) but as commented by the authors, subsequent work to characterize the enzyme from strain QB4 is currently ongoing. Since the contents of the results and discussion are based on the characteristics of the enzymes which is degradation of uronic acid, I still feel that it is necessary for the authors to provide data to at least show that the growth of the cells is due to the utilization of the substrate. Cell growth does not fully justify that the polysaccharides are being utilized or degraded. Please comment or provide the necessary data.

Besides the point above, I see that sufficient data was provided to further justify their claims in this work.

Experimental design

No further comments.

Validity of the findings

As commented in the "Basic reporting" section, the authors just need to justify that strain QB4 does utilize the designated polysaccharides (Fig.1 and Fig. 2).

Additional comments

No further comments

---

## Round 0.3 · Minor Revisions

I am mostly satisfied with your replies, but I must ask you to adjust the (newly-added) paragraph encompassing lines 460-478: "It was known that E. chrysanthemi also possessed the polygalacturonate pathway and the isomerase pathway. Even though the enzyme production of the isomerase pathway was not impaired by mutation in gens of the polygalacturonate pathway (kduD and kduI), these mutants were inhibited their growth on polygalacturonated (Hugouvieux-Cotte-Pattat et al., 1996). In the KEGG pathway database, it was found that Dicleya Dadantii belonging to the Gammaproteobacteria as well as E. chrysanthemi and QB4 also possesses the polygalacturonate pathway and the isomerase pathway. However, the transcriptomic analysis of D. Dadantii during the early colonization on the plant leaf demonstrated that genes involved in the polygalacturonate pathway were upregulated in the condition (Pédron et al., 2018). These results suggested that the isomerase pathway was not a main pathway for utilizing pectin in Gammaproteobacteria. Third pathway, epimerase pathway, involving the conversion of D-tagatose to D-fructuronate by tagatose epimerase (UxaE) was found in the hyperthermophilic bacterium, Thermotoga maritima, belonging to the phylum Thermotoga. However, the pathway was not found in other bacteria, such as Escherichia and Bacillu (Kuivanen et al., 2019). uxaE and other epimerase pathway’s genes, such as hexuronate catabolism regulator (uxaR), fructuronate reductase (uxaD), predicted D-mannonate utilization enzyme (gntE) were colocalized in the chromosome of the genus Thermotoga (Rodionova et al., 2012). However, Table 2 demonstrated that genes encoding Pu14, Pu15, and Pu16 were scatteredly distributed in the chromosome and plasmid. This localization suggested that the pathway in QB4 might not function."

Specifically:

1) The paragraph appears disconnected from the previous sentences, and it is not immediately apparent why you start speaking of E. chrysanthemi. I suggest you begin the paragraph with an introductory sentence stating the objective of the paragraph itself (e.g. " Several pathways for xx utilization have been described in Bacteria", or something like that.

2) E. chrysanthemi is a synonym of Dickeya dadantii DOI:10.1099/ijs.0.02791-0

3) There are several typos and grammatical errors throughout the paragraph (e.g. "Bacillu" instend of "Bacillus", etc.)

4) The final sentence is cryptic: "This localization suggested that the pathway in QB4 might not function." Which pathway is this sentence referring to? Also, how why do you make such a conclusion from the gene distribution in the chromosome/plasmid ? The reasoning should be made clear.

---

## Round 0.4 · accepted · Accept

I am glad to accept your paper.Congratulations!

Reviewer 1 ·

Basic reporting

No comment

Experimental design

No comment

Validity of the findings

No comment

Additional comments

The authors have improved the article according to reviewers recommendations. In my opinion it is ready for publication.

Reviewer 3 ·

Basic reporting

No further comments

Experimental design

No further comments

Validity of the findings

No further comments

Additional comments

No further comments